# Adhesion Strength Enhancement of Butyl Rubber and Aluminum Using Nanoscale Self-Assembled Monolayers of Various Silane Coupling Agents for Vibration Damping Plates

**DOI:** 10.3390/nano14181480

**Published:** 2024-09-11

**Authors:** So Rim Lee, Dang Xuan Nghia, Jin Young Oh, Tae Il Lee

**Affiliations:** 1Department of Materials Science and Engineering, Gachon University, Seong-nam 13306, Gyeonggi, Republic of Korea; isrollo@gachon.ac.kr (S.R.L.); 314159@gachon.ac.kr (D.X.N.); 2Department of Chemical Engineering (Integrated Engineering Program), Kyung Hee University, Yongin 17104, Gyeonggi, Republic of Korea

**Keywords:** aluminum, butyl rubber, silane coupling agent, self-assembly monolayer

## Abstract

In this paper, we enhance the adhesion strength of butyl rubber-based vibrational damping plates using nanoscale self-assembled monolayers of various silane coupling agents. The silane coupling agents used to chemically modify the plate’s aluminum surface include 3-aminopropyltriethoxysilane (APTES), (3-glycidyloxypropyl) triethoxysilane (GPTES), 3-mercaptopropyltrimethoxysilane (MPTMS), and 3-(triethoxysilyl)propyl isocyanate (ICPTES). The modified surfaces were analyzed using Fourier transform infrared spectroscopy (FT-IR) and X-ray photoelectron spectroscopy (XPS), and the enhancement in adhesion strength between the rubber and aluminum was estimated through T-Peel tests. As a result, MPTMS showed the highest enhancement in adhesion strength, of approximately 220% compared to the untreated sample, while GPTES, ICPTES, and APTES resulted in adhesion strength enhancements of approximately 200%, 150%, and 130%, respectively.

## 1. Introduction

The mechanical vibration generated from the operation of auto vehicle engines and motors is transmitted to the interior of the vehicle, decreasing ride comfort and making it difficult to drive [1]. To solve this problem, vibration damping plates (VDPs), which convert vibrational energy into thermal energy, are widely used in automotive parts [2,3,4]. VDPs are often designed to have a sandwich structure consisting of metal/viscoelasticity resin/metal, which takes into account their manufacturing processability. Recently, aluminum has replaced steel in VDPs to reduce the automobile’s weight for fuel efficiency. It also offers advantages such as good machinability, cold resistance, high thermal conductivity, and a low cost, making it an ideal material for electric vehicle case components [5,6,7,8,9].

However, aluminum forms weak physical bonds with polymeric viscoelastic resin, leading to a low VDP adhesion strength and eventual peeling issues, which limit its pressing process and application [10,11]. Therefore, enhancing the adhesion strength between this resin and aluminum through surface modification is essential. One effective method involves using silane coupling agents, which are known to improve adhesion by forming covalent bonds between polymer and metal surfaces [12,13,14,15]. In our previous research, we boosted the adhesion strength between butyl rubber and aluminum through the use of 3-mercaptopropyltrimethoxysilane (MPTMS) as a coupling agent; this led to the formation of covalent bonds between the alkene groups in butyl rubber and the thiol groups in MPTMS via thiol-ene reactions [16]. However, to date, there has been limited exploration of the use of silane-based agents beyond MPTMS to enhance the adhesion strength of metal–elastomer-based damping plates [17,18].

In this study, we achieved adhesion enhancement through the use of silane coupling agents with various functional groups, which were used in the form of self-assembled monolayers; these are one-molecule-thick layers of material that form an organized structure, binding to a surface due to physical or chemical processes. Specifically, we applied these coupling agents to butyl rubber and aluminum plate-based VDPs. Silane coupling agents have a silane group capable of forming covalent bonds with the aluminum’s surface and a functional group that forms covalent bonds with the resin, resulting in an enhancement of adhesion strength [19,20,21,22]. The silane coupling agents used in this work were 3-aminopropyltriethoxysilane (APTES), (3-glycidyloxypropyl)triethoxysilane (GPTES), 3-mercaptopropyltrimethoxysilane (MPTMS), and 3-(triethoxysilyl)propyl isocyanate (ICPTES). These agents form Si−O−Al bonds with the aluminum plates and possess different functional groups that enable their covalent bonding with the butyl rubber resin: APTES contains an amino group (−NH₂), GPTES has an epoxy group (−CH(O)CH₂), MPTMS contains a thiol group (−SH), and ICPTES includes an isocyanate group (−N=C=O). The structure and chemical composition of the silane coupling agents used to coat the plates’ aluminum surface were verified using Fourier transform infrared spectroscopy (FT-IR), X-ray photoelectron spectroscopy (XPS), and contact angle measurements. The adhesion strength of the VDPs containing different silane coupling agents was estimated through T-Peel strength tests.

## 2. Materials and Method

Aluminum plates (0.2 × 20 × 80 mm) were cleaned using acetone, isopropyl alcohol, and DI water to remove contaminants from the surface. Subsequently, their surface was etched with a 1 wt % aqueous solution of NaOH for 10 min to form hydroxyl groups terminated at the aluminum surface. Then, the aluminum plates were sonicated in deionized (DI) water for 10 min. After that, they were placed in silane coupling agent solutions for 7 min. We combined 1 mL of silane coupling agent, 2.5 mL of DI water, and 46.5 mL of ethanol to create the solution (silane coupling agents: APTES, GPTES, MPTMS, ICPTES, all 95%, Sigma Aldrich, Burlington, MA, USA). After the silane coupling agent treatment, the aluminum plates were cleaned with a 95% ethanol + 5% deionized (DI) water solution via two 10 min rounds of sonication, dried with 99.99% nitrogen gas, and annealed at 100 °C on a hot plate. Finally, a wire bar coater (RDS 65) was used to apply butyl rubber resin to the aluminum plates, and then the plates were heated to 160 °C for 5 min. To complete the VDPs, the resin-coated surfaces of two aluminum plates were then bonded together at a speed of 1 at 160 °C using a laminator (Prime-470D).

The adhesion strength of the VDPs was estimated using a T-Peel test, which was conducted on a universal testing machine MCT-2150 at a T-Peel speed of 10 mm/min. Using a K-alpha spectrometer (Thermo VG, Cambridge, UK) with monochromated Al X-ray sources (Al Kα line: 1486.6 eV), the chemical composition of both treated and untreated aluminum surfaces was examined. The XPS analysis was performed at an X-ray power of 2 kV/3 mA. Using a syringe filled with 3 mL of DI water, the water contact angle of the surface of aluminum was measured using an SEO-Phoenix 10 contact angle meter. Additionally, the existence of silane coupling agents within the plates was observed through FT-IR (Vertex 70) analysis through a service provided by the Center for Bionano Materials Research at Gachon University.

## 3. Results

### 3.1. Water Contact Angle

The contact angle change of aluminum surfaces treated with various silane coupling agents is depicted in Figure 1. Untreated aluminum has a contact angle of 77.87°. This number shows that Al−OH and Al−O−Al bonds are present on the cleaned aluminum surface, which means it is highly hydrophobic due to a lack of polar functional groups. When treated with NaOH, the contact angle of the aluminum surface decreases to 13.76°, suggesting that this etching process breaks the Al−O−Al bonds and increases the density of OH group terminations (Al−OH) on the surface. When the aluminum plate is submerged in a solution containing a silane coupling agent, the hydroxyl groups that were produced on the aluminum surface after the NaOH treatment react strongly with the silane groups in APTES, GPTES, MPTMS, and ICPTES, forming Si−O−Al bonds. Finally, the aluminum surface is terminated with the functional groups of APTES, GPTES, MPTMS, and ICPTES, which are an amino group, epoxy group, thiol group, and isocyanate group, respectively. Since all of these functional groups have a lower dipole moment than the hydroxyl group, the contact angles of these aluminum plates increase slightly after the NaOH treatment, to 17.99°, 21.73°, 38.03°, and 20.61°, respectively. Lower water contact angles are generally associated with higher surface energies, as surface energy increases with the polarity of the exposed functional groups. Figure 1 illustrates that the polarity strengths of the functional groups are ranked as follows: hydroxyl, amine, isocyanate, epoxy, and thiol. By comparing the changes in contact angle following the surface treatment process, it was confirmed that self-assembled monolayers of APTES, GPTES, MPTMS, and ICPTES were successfully formed on the aluminum plate’s surface.

### 3.2. FT-IR Analysis

The existence of a silane coupling agent on the surface of the treated aluminum plates was determined using FT-IR analysis. Figure 2a shows the absorbance of the APTES-treated aluminum surface. The peaks at 3444 ${cm}^{-1}$ and 1674 ${cm}^{-1}$ correspond to a N−H stretching vibration and ${NH}_{2}$ bending, respectively. The 3020 ${cm}^{-1}$ peak indicates CH stretching vibration, while the 2376 peak arises from ${CO}_{2}$ absorption [23]. The 1535 ${cm}^{-1}$ peak is due to the bending of absorbed water [24]. The peaks at 1204 ${cm}^{-1}$ and 1107 ${cm}^{-1}$ represent Si−C symmetric bending and Si−O and Al−O asymmetric bending, respectively, while the peak at 930 ${cm}^{-1}$ corresponds to Si−CH stretching. Figure 2b shows the absorbance of the GPTES-treated aluminum surface. The 2922 ${cm}^{-1}$ peak comes from C−H stretching, while the 2345 ${cm}^{-1}$ peak comes from ${CO}_{2}$ absorption [25]. The peak at 1456 ${cm}^{-1}$ is due to the bending vibrations of ${CH}_{2}$ and ${CH}_{3}$. The peak at 1163 ${cm}^{-1}$ is associated with the stretching vibration of the C−O group, and the 812 ${cm}^{-1}$ and 701 ${cm}^{-1}$ peaks represent the asymmetric bending of Si−O and Al−O and the bending of $H_{2}O$, respectively [26]. Figure 2c shows the absorbance of the MPTMS-treated sample. The peak near 3367 ${cm}^{-1}$ is due to the OH stretching vibration in MPTMS. The 2931 ${cm}^{-1}$ peak is due to CH stretching vibration, and the 2559 ${cm}^{-1}$ peak indicates the presence of -SH groups. The 2366 ${cm}^{-1}$ peak indicates ${CO}_{2}$ absorption. The peaks at 1630 ${cm}^{-1}$ and 759 ${cm}^{-1}$ are caused by the bending of $H_{2}O$, while the 1454 ${cm}^{-1}$ peak is due to ${CH}_{2}$ and ${CH}_{3}$ vibrational bending [27]. The peaks at 1255 ${cm}^{-1}$ and 1074 ${cm}^{-1}$ correspond to Si−C symmetric bending and Si−O and Al−O asymmetric stretching, respectively, and the 854 ${cm}^{-1}$ peak is due to Si−CH stretching. Finally, Figure 2d shows the absorbance of the ICPTES-treated aluminum surface. The peak at 2382 ${cm}^{-1}$ represents ${CO}_{2}$ absorption, while the peak at 2341 ${cm}^{-1}$ indicates an isocyanate double bond [28,29]. The 1581 ${cm}^{-1}$ and 1286 ${cm}^{-1}$ peaks correspond to ${CH}_{2}$ and ${CH}_{3}$ bending vibrations and Si−CH stretching, respectively, while the asymmetric stretching of Si−O and Al−O in the Si−O−Al and Al−O−H bonds is responsible for the peak at 1134 ${cm}^{-1}$ [30,31]. These results confirm that self-assembled monolayers of the silane coupling agents were successfully formed on the aluminum plates’ surface.

### 3.3. XPS Analysis

The composition and chemical bonding state of the aluminum surfaces after treatment with silane coupling agents were investigated using the XPS spectra shown in Figure 3. The atomic concentrations calculated from these spectra are listed in Table 1. In terms of the APTES-treated aluminum, peaks corresponding to N1s (401 eV), Si2s (153.9 eV), and Si2p (102.1 eV) were obtained from its surface. From this analysis, it was confirmed that the APTES-treated aluminum plate’s surface contained the elements C, Si, O, Al, and N at atomic concentrations of 33.64% (C1s), 2.65% (Si2p), 46.5% (O1s), 14.83% (Al2p), and 2.38% (N1s) [32]. For the GPTES-treated aluminum, peaks corresponding to O1s (531.7 eV), C1s (285.22 eV), Si2p (154.3 eV), and Si2p (100.6 eV) were observed. This verified that the aluminum surface contained the components C, Si, O, and Al at concentrations of 29.09% (C1s), 1.31% (Si2p), 49.42% (O1s), and 20.18% (Al2p), respectively [33,34]. Peaks corresponding to S2p (163.5 eV), S2s (228.0 eV), Si2s (152.0 eV), and Si2p (101.8 eV) were detected for the aluminum surface treated with MPTMS. These elements’ atomic concentrations were found to be 29.2% (C1s), 3.1% (Si2p), 5.5% (S2p), 17% (Al2p), and 45.1% (O1s) [35]. Finally, for the ICPTES-treated aluminum, peaks at O1s (531 eV), N1s (401.3 eV), C1s (284.7 eV), Si2s (151.9 eV), and Si2p (100.7 eV) were observed on the surface. This demonstrated that, following ICPTES treatment, C, Si, O, Al, and N were present on the surface at atomic concentrations of 30.01% (C1s), 1.64% (Si2p), 47.6% (O1s), 19.46% (Al2p), and 1.29% (N1s). To confirm whether functional groups were stably formed on the surface of these plates, the presence of the characteristic bonds of the functional groups was verified using high-resolution XPS, as shown in Figure 4. Figure 4b presents the deconvolution of the N1 peak of APTES-treated aluminum, revealing that nitrogen is present and that approximately 60.78% of it is in N−C (400.21 eV) and 39.22% is in N−H (401.42 eV) bonds. Figure 4c shows the deconvolution of the C1s peak of GPTES-treated aluminum, indicating that carbon is present, with about 69.19% of it in C−C (284.42 eV) and 30.81% in C−O−C (286.02 eV) bonds. The S2p peak deconvolution of MPTMS-treated aluminum is shown in Figure 4d, confirming that sulfur is present and that 51.1% of it is $-S_{x}OH$ (163.54 eV) and 48.0% is −SS and −SH (167.01 eV). Finally, Figure 4e shows the deconvolution of the N1s peak of ICPTES-treated aluminum, indicating that the nitrogen present consists of approximately 10.45% N−C (399.28 eV) and 89.55% N=C (400.41 eV) bonds [36]. These results lead to the conclusion that the Si−OH groups of the silane coupling agents react with the OH groups on the NaOH-treated aluminum surface to form self-assembled silane coupling agent monolayers.

### 3.4. Adhesion Strength Test

A T-Peel test was conducted to measure the change in the adhesion strength between aluminum and butyl rubber depending on the type of silane coupling agent used. Figure 5a shows representative peel strength curves for plates treated with NaOH, APTES, GPTES, MPTMS, and ICPTES, respectively. The results indicate that the peel strength increased with self-assembled monolayer formation, regardless of the type of silane coupling agent used. The adhesion strength between the plate and the resin was improved by the bonding of the self-assembled monolayer to both the aluminum surface and the butyl rubber. Figure 5b shows representative peel strength curves and the average peel strength of each type of silane coupling agent. Of these, the MPTMS self-assembled monolayer showed the highest adhesive strength, with a peel strength increase of up to 220% compared to the untreated aluminum. GPTES was the next strongest, with a peel strength increase of up to 200%, then ICPTES with 150%, and APTES with 130%.

The mechanism by which the silane coupling agent treatment enhances the VDPs’ peel strength is illustrated in Figure 6. For all samples, the NaOH treatment induces the formation of OH groups on the aluminum’s surface. Figure 6a shows the mechanism of the APTES treatment. In an aqueous solution, APTES undergoes hydrolysis, converting its ${Si-OCH}_{2}{CH}_{3}$ group into Si−OH. The converted Si−OH groups then interact with the OH groups on the aluminum’s surface, forming hydrogen bonds. Through an annealing process, these hydrogen bonds undergo dehydration to form Si−O−Al bonds. As a result, the aluminum surface becomes terminated with $NH_{2}$ groups. When the surface-treated aluminum is coated with butyl rubber and subjected to annealing at 160 °C, bonds such as C−NH and C−H are formed due to the interaction between the C=C group in the butyl rubber and the $NH_{2}$ groups on the aluminum’s surface. The C−NH bond serves as a covalent bonding bridge in this process, connecting the aluminum surface with the butyl rubber. Figure 6b illustrates the mechanism of the GPTES treatment; similar to the APTES process, GPTES reacts with the OH groups on the aluminum surface, forming a $CH(O){CH}_{2}$-terminated self-assembly monolayer. After that, the aluminum surface is coated with butyl rubber and subjected to annealing. During this process, the C=C groups in the butyl rubber react with the $-CH(O){CH}_{2}$ groups of the GPTES, leading to the formation of ${C-C\left(O\right)-CH}_{2}$ and C–H bonds. As a result, the ${C-C\left(O\right)-CH}_{2}$ acts as a covalent link connecting the metal surface and the butyl rubber. Figure 6c displays the mechanism behind the MPTMS treatment. Following the same process as before, the aluminum surface was treated with NaOH, which facilitated the formation of an SH-terminated self-assembled monolayer on the surface. After coating the surface with butyl rubber and annealing the plate, a thiol-ene reaction occurs between the C=C groups in the butyl rubber and the −SH groups in the MPTMS. As a result of this reaction, C−S and C−H bonds are formed. The C−S bond serves as a covalent connecting bridge between the surface of the aluminum and the butyl rubber. Figure 6d illustrates the mechanism behind the ICPTES treatment. In an aqueous solution, ICPTES undergoes hydrolysis, converting its ${Si-OCH}_{2}{CH}_{3}$ groups into Si−OH, while its functional N=C=O groups are converted into NH−C(O)−OH. As IPTES is combined with the OH on the aluminum’s surface, a NH−C(O)−OH-terminated self-assembled monolayer is produced. During the subsequent butyl rubber coating and annealing process, the C=C groups in the butyl rubber react with the NH−C(O)−OH groups, leading to the formation of C−O−C(O)−NH and C−H bonds; C−O−C(O)−NH serves as a covalent connecting bridge between the aluminum surface and butyl rubber. Upon examining the mechanisms at work in the reaction of each type of silane coupling agent, it was determined that the covalent connection they produced enhanced the peel strength between the aluminum and butyl rubber.

Among these agents, the −SH group in MPTMS formed the strongest bonds with the C=C groups in the butyl rubber through a thiol-ene reaction. The thiol-ene reaction is a process widely used in fields such as coatings, adhesives, and energy damping due to its high efficiency, rapid reaction rate, high conversion rate, and low shrinkage [37,38,39]. These advantageous features likely contribute to the significant improvement in peel strength observed when MPTMS is used as a surface treatment, making it the most effective among the silane coupling agents studied.

## 4. Conclusions

In this study, nanoscale self-assembled monolayers of silane coupling agents were used to enhance the adhesion strength between aluminum and butyl rubber. After creating OH groups on the aluminum’s surface via a NaOH treatment, self-assembled monolayers were formed using APTES, GPTES, MPTMS, and ICPTES. To assess the qualitative characteristics of the treated aluminum surfaces and the changes in their chemical compositions, FT-IR and XPS analyses were conducted, and the results confirm that the silane coupling agents successfully bonded to the aluminum surface, forming self-assembled monolayers. Furthermore, the T-Peel test was conducted to measure the changes in adhesion strength observed depending on the type of silane coupling agent used. The results show that the MPTMS treatment achieved the greatest increase in adhesion strength, of approximately 220% compared to the untreated aluminum plate, which was followed by GPTES, with about a 200% increase, ICPTES, with about a 150% increase, and APTES, with about a 130% increase in adhesion strength. The observed increase in adhesion strength resulted from the formation of covalent bonds between the silane coupling agents’ functional groups and the double-bonded carbon atoms in the butyl rubber. It is believed that, under the specified process conditions, the variations in adhesion strength enhancements observed are due to the kinetic differences between the various bonding reactions of these coupling agents. Of these reactions, the thiol-ene reaction is considered the most effective for forming covalent bonds.

## Figures and Tables

**Figure 1 nanomaterials-14-01480-f001:**
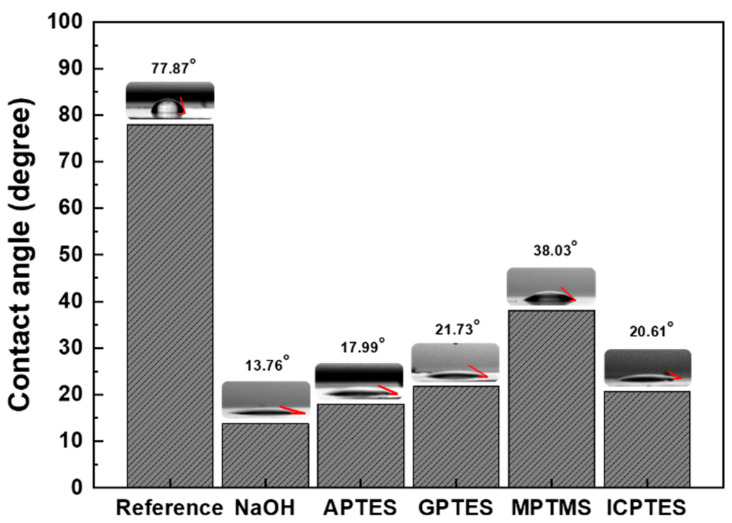
Water contact angle of untreated, NaOH-treated, APTES-treated, GPTES-treated, MPTMS-treated, and ICPTES-treated aluminum surfaces.

**Figure 2 nanomaterials-14-01480-f002:**
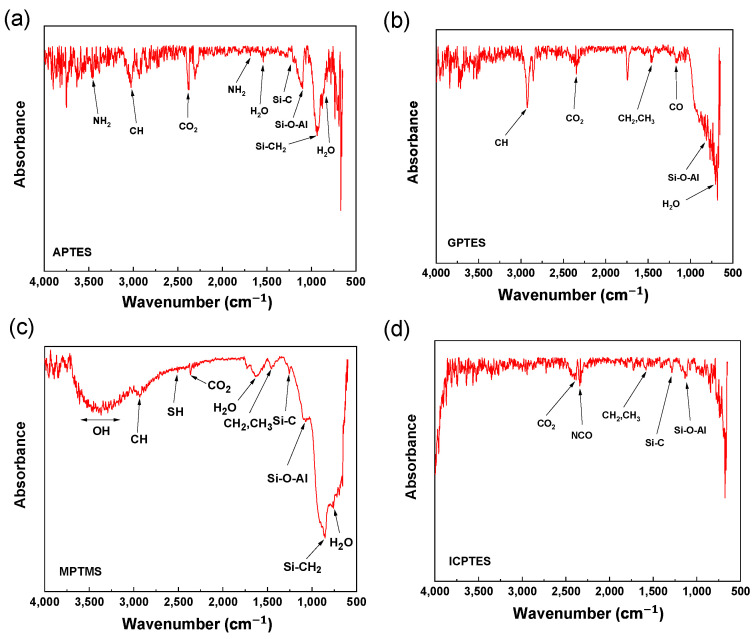
FT-IR spectra of (**a**) APTES-treated, (**b**) GPTES-treated, (**c**) MPTMS-treated, and (**d**) ICPTES-treated aluminum surfaces.

**Figure 3 nanomaterials-14-01480-f003:**
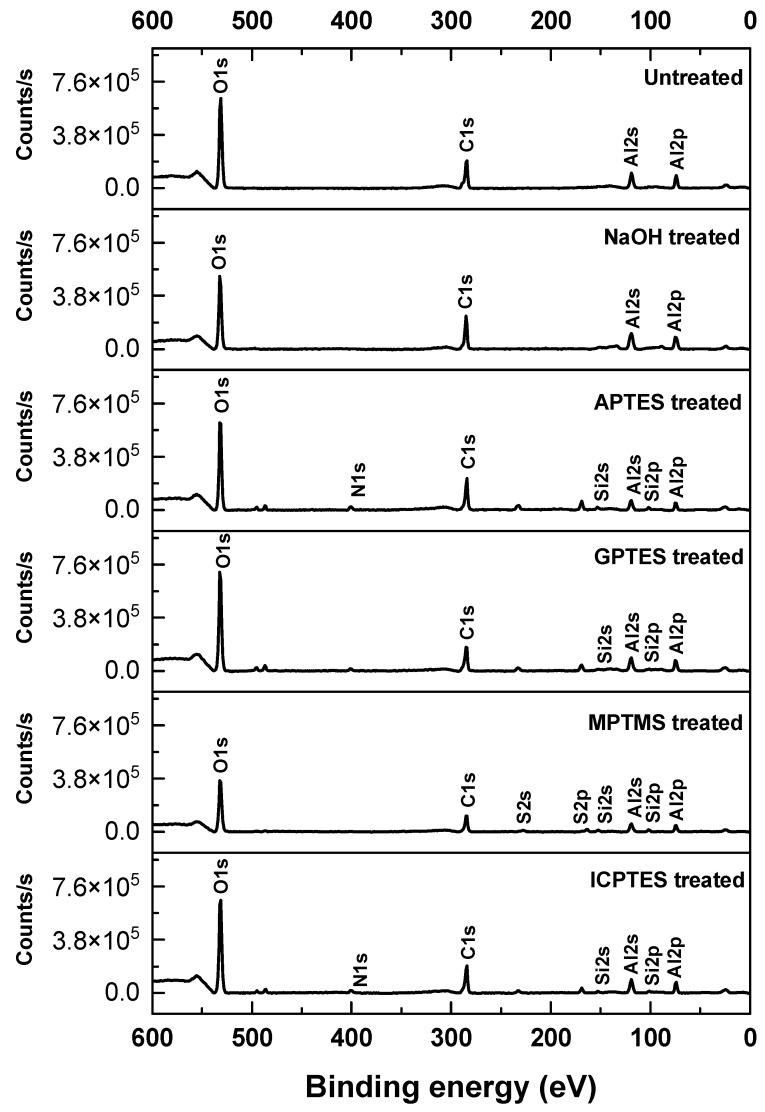
XPS spectra of the untreated, NaOH-treated, APTES-treated, GPTES-treated, MPTMS-treated, and ICPTES-treated aluminum plates.

**Figure 4 nanomaterials-14-01480-f004:**
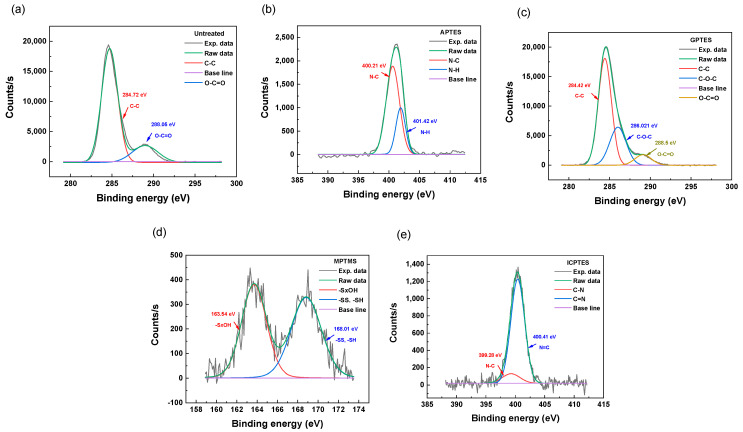
High-resolution deconvoluted XPS spectra of (**a**) C1s of untreated Al, (**b**) N1s of APTES, (**c**) C1s of GPTES, (**d**) S2p of MPTMS, and (**e**) N1s of ICPTES.

**Figure 5 nanomaterials-14-01480-f005:**
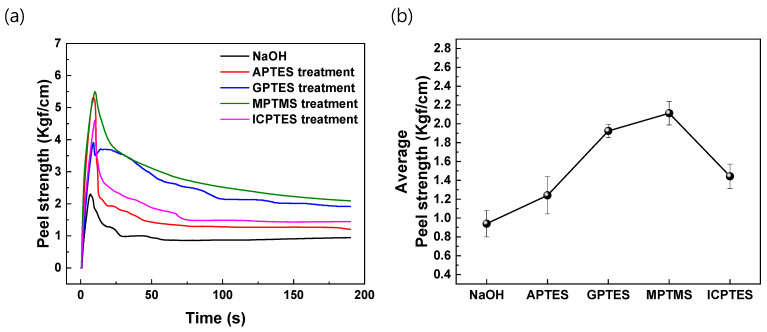
Comparison of the peel strength curve of NaOH with the peel strength curves of different types of silanes coupling agents (**a**) and the average peel strength of each type of silane coupling agent (**b**).

**Figure 6 nanomaterials-14-01480-f006:**
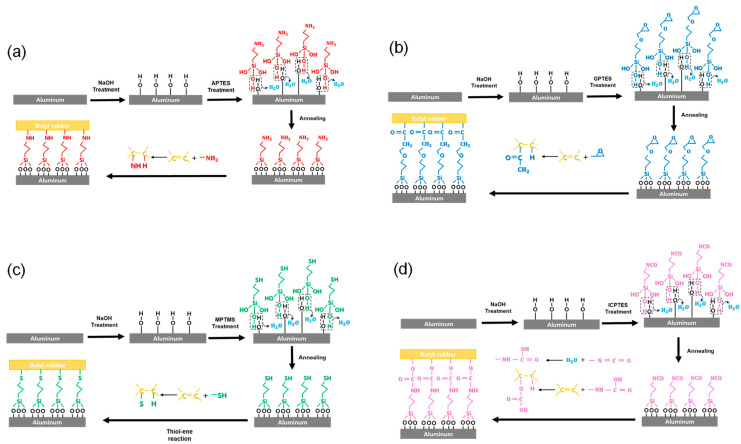
A step-by-step illustration of the enhancement of the adhesion force between aluminum and butyl rubber through an (**a**) APTES treatment, (**b**) GPTES treatment, (**c**) MPTMS treatment, and (**d**) ICPTES treatment.

**Table 1 nanomaterials-14-01480-t001:** Atomic percentages of different elements in untreated, NaOH-treated, APTES-treated, GPTES-treated, MPTMS-treated, and ICPTES-treated aluminum surfaces, obtained from XPS analysis.

Elements	C	Si	O	Al	N	S
Untreated	29.8	-	50.0	20.2	-	-
NaOH	30.6	-	44.0	25.4	-	-
APTES	33.64	2.65	46.5	14.83	2.38	-
GPTES	29.09	1.31	49.42	20.18	-	-
MPTMS	29.2	3.1	45.1	17	-	5.5
ICPTES	30.01	1.64	47.6	19.46	1.29	-

## Data Availability

The original contributions presented in the study are included in the article, further inquiries can be directed to the corresponding authors.

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
