# Peer review of "Adhesion Strength Enhancement of Butyl Rubber and Aluminum Using Nanoscale Self-Assembled Monolayers of Various Silane Coupling Agents for Vibration Damping Plates"

_nanomaterials, 2024, doi:10.3390/nano14181480_

Round 1

Reviewer 1 Report

Comments and Suggestions for Authors

This manuscript reports the enhancement of adhesion strength between butyl rubber and aluminum using various silane coupling agents, which might have promising application in vibration damping plates. The study is basically comprehensive and the finding is interesting. However, the discussions are insufficient and not deep enough, which requires major revision. The manuscript might be accepted for publication after addressing the following issues:

(1) The author has not discussed the reason for the different contact angles of surfaces treated with different silane coupling agents.

(2) In the FT-IR and XPS analysis part, the author only confirms the successful self-assembly of silane coupling agents on surfaces. But what is the grafting density of different surfaces? This might be crucial for the explanation of the different adhesion strength enhancement.

(3) The peel strength result should include the variation of the data. Error bars should be given in the figures.

(4) The reason for the different enhancement of adhesion strength of different surfaces should be provided in the conclusion.

Author Response

Review1’s Comments and Suggestions for Authors

This manuscript reports the enhancement of adhesion strength between butyl rubber and aluminum using various silane coupling agents, which might have promising application in vibration damping plates. The study is basically comprehensive and the finding is interesting. However, the discussions are insufficient and not deep enough, which requires major revision. The manuscript might be accepted for publication after addressing the following issues:

(1) The author has not discussed the reason for the different contact angles of surfaces treated with different silane coupling agents.

Our response: Thank you for the constructive comment. Lower water contact angles are generally associated with higher surface energies, as surface energy increases with the polarity of the exposed functional groups. Figure 1 illustrates that the polarity strengths of the functional groups are ordered as follows: hydroxyl, amine, isocyanate, epoxy, and thiol. This content has been added in revised the text.

Page 3 line 38~42

Lower water contact angles are generally associated with higher surface energies, as surface energy increases with the polarity of the exposed functional groups. Figure 1 illustrates that the polarity strengths of the functional groups are ordered as follows: hydroxyl, amine, isocyanate, epoxy, and thiol.”

(2) In the FT-IR and XPS analysis part, the author only confirms the successful self-assembly of silane coupling agents on surfaces. But what is the grafting density of different surfaces? This might be crucial for the explanation of the different adhesion strength enhancement.

Our response: We agree with the reviewer's comments. It is true that one factor that may influence the adhesion strength is the density of the self-assembled monolayer (SAM) formed by the silane coupling agent. The density of SAM can be quantitatively measured using techniques like angle-resolved XPS (X-ray photoelectron spectroscopy) and AFM (atomic force microscopy). However, since commercially available aluminum sheets were employed in this study, the surface roughness is considerable, making it difficult to conduct such precise measurements to determine the density of the nanometer-thick SAM layer. Therefore, we used silane coupling agents with comparable propyl carbon chain lengths and carried out SAM formation under identical processing conditions to ensure the creation of SAM layers with as uniform a density as possible.

(3) The peel strength result should include the variation of the data. Error bars should be given in the figures.

Our response: To address the reviewer's comment, error bars have been incorporated into the updated Figure 5(f).

Figure R1. The revised Figure 5(f) with error bars

(4) The reason for the different enhancement of adhesion strength of different surfaces should be provided in the conclusion.

Our response: Thank you for your comment. we have added the following sentences in the conclusion of revised manuscript.

Page 6 line 53~ Page 7 line 6

 “ The observed increase in adhesion strength resulted from the formation of covalent bonds between the silane coupling agents’ functional groups and the double-bonded carbon atoms in the butyl rubber. It is believed that, under the specified process conditions, the variations in adhesion strength enhancements observed are due to the kinetic differences between the various bonding reactions of these coupling agents. Of these reactions, the thiol-ene reaction is considered the most effective for forming covalent bonds.”

Reviewer 2 Report

Comments and Suggestions for Authors

The manuscript titled “Adhesion strength enhancement of butyl rubber and aluminum using various silane coupling agents for vibration damping plates” is presenting a comparison between different silane coupling agents, used as adhesion improvement solutions, in terms of chemistry and adhesion behavior.

This paper is recommended for publication in the Nanomaterials journal, after minor revisions.

Following are only some of the issues in the manuscript which are to be seriously addressed for accepting its publication in the journal:

1.       The English language and grammar used in the present manuscript needs significant improvement. There are plenty of instances where mistakes, misspelled words and/or poorly chosen words (ambiguous), and unnecessary repetitions are present. I strongly suggest that the paper should be proofread and double-checked concerning the spelling and phrasing. This version is relatively difficult to read and to understand.

2.       Since the manuscript was submitted to Nanomaterials, the “nano” aspect should be emphasized.

3.       “However, to date, there has been limited exploration of silane-based agents beyond MPTMS for enhancing the adhesion between butyl rubber and aluminum for damping plates.” “Limited” means that there are some reports about this subject. References should be added./span/p p class="MsoListParagraphCxSpMiddle" style="text-indent: -.25in; mso-list: l0 level1 lfo1"4.       Correct the details related to the bonding process.

5.       It is not clear why those particular spectra from figure 4 were chosen to be shown for each sample configuration.

6.       The “self-assembled monolayer” wording should be defined for this manuscript.

7.       The comparisons from figure 5 (a-d) should be removed, since they are presented again in figure 5e.

Comments on the Quality of English Language

see report

Author Response

Review2’s Comments and Suggestions for Authors

The manuscript titled “Adhesion strength enhancement of butyl rubber and aluminum using various silane coupling agents for vibration damping plates” is presenting a comparison between different silane coupling agents, used as adhesion improvement solutions, in terms of chemistry and adhesion behavior.

This paper is recommended for publication in the Nanomaterials journal, after minor revisions.

Following are only some of the issues in the manuscript which are to be seriously addressed for accepting its publication in the journal:

(1) The English language and grammar used in the present manuscript needs significant improvement. There are plenty of instances where mistakes, misspelled words and/or poorly chosen words (ambiguous), and unnecessary repetitions are present. I strongly suggest that the paper should be proofread and double-checked concerning the spelling and phrasing. This version is relatively difficult to read and to understand.

Our response: Thank you for your comment. Our manuscript has been edited in English through the institutional service provided by MDPI

(2) Since the manuscript was submitted to Nanomaterials, the “nano” aspect should be emphasized.

 Our response: Thank you for your comment. Enhancing adhesion strength through the formation of self-assembled monolayers of silane coupling agents with nanometer thicknesses is the primary objective of this work. In the title, abstract, introduction, and conclusion sections, we emphasized the development of nanoscale crosslinking layers that facilitate covalent bonding between aluminum and butyl rubber.

(3) “However, to date, there has been limited exploration of silane-based agents beyond MPTMS for enhancing the adhesion between butyl rubber and aluminum for damping plates.” “Limited” means that there are some reports about this subject. References should be added./span/p p class="MsoListParagraphCxSpMiddle" style="text-indent: -.25in; mso-list: l0 level1 lfo1"

Our response: Thank you for your comment. We have added the related references after the sentence.

(17) Sang,J. ; Aisawa, S.; Miura, K.; Hirahara, H.; Jan, O.; Jozef, P.; Pavol, M. Adhesion of carbon steel and natural rubber by functionalized silane coupling agents. International Journal of Adhesion and Adhesives 2017, 72,70-74

(18) Jayaseelan, S. K.; Van Ooij W. J. Rubber-to-metal bonding by silanes. Journal of Adhesion Science and Technology 2001,15(8) 967-991

(4)  Correct the details related to the bonding process.

Our response: Thank you for your comment. We have revised the missing details in the reaction between silane and hydroxyl groups in the bond formation process shown in Figure 6.

Figure R2. The revised Figure 6

(5)  It is not clear why those particular spectra from figure 4 were chosen to be shown for each sample configuration.

Our response: Thank you for your comment. To verify that each functional group is stably present on the surface, it was necessary to confirm the presence of characteristic bonds for each functional group. To clarify the meaning of Figure 4, we have added related explanations into the main text.

Page 5 line 1 ~ 3

“To confirm whether functional groups were stably formed on the surface of these plates, the presence of the characteristic bonds of the functional groups was verified using high-resolution XPS, as shown in Figure 4.”

(6) The “self-assembled monolayer” wording should be defined for this manuscript.

Our response: Thank you for your comment. We have added an explanation of what a self-assembled monolayer is to the introduction section.

Page 2 line 17 ~20

“In this study, we obtained adhesion enhancement through the use of silane coupling agents with various functional groups, which were used in the form of self-assembled monolayers; these are one-molecule-thick layers of material that form an organized structure, binding to a surface due to physical or chemical processes. Specifically, we applied these coupling agents to butyl rubber and aluminum plate-based VDPs.”

(7) The comparisons from figure 5 (a-d) should be removed, since they are presented again in figure 5e.

Our response: Thank you for your comment. We agree with the reviewer's opinion. Figure 5 has been revised as follows:

Figure R3. The revised Figure5

Round 2

Reviewer 1 Report

Comments and Suggestions for Authors

The author has basically addressed the major conerns, which makes it appropriate for publication in this journal.

Author Response

Comment1: The author has basically addressed the major conerns, which makes it appropriate for publication in this journal.

Response: Thank you for your comment.